# Biomechanical Effects of a Passive Back-Support Exosuit During Simulated Military Lifting Tasks—An EMG Study

**DOI:** 10.3390/s25103211

**Published:** 2025-05-20

**Authors:** Muhammad Ammar Marican, Lavern Dharma Chandra, Yunqi Tang, Muhammad Nur Shahril Iskandar, Cheryl Xue Er Lim, Pui Wah Kong

**Affiliations:** 1Physical Education and Sports Science Department, National Institute of Education, Nanyang Technological University, Singapore 637616, Singapore; 2College of Art and Design, Shaanxi University of Science and Technology, Xi’an 710021, China; 3Independent Researcher, Singapore 126755, Singapore

**Keywords:** Auxivo, electromyography, exoskeleton, muscle activity, spine, load

## Abstract

Military operators performing vehicle maintenance work are at times subject to onerous tasks such as lifting and transporting heavy loads, potentially in confined spaces. As this presents a risk for developing musculoskeletal injury, it is of interest to evaluate if a passive back-support exosuit could help reduce back muscle load. This study used wireless electromyographic (EMG) sensors to evaluate the biomechanical effects of exosuits during lifting tasks. Ten male participants performed military-relevant lifting tasks with and without wearing the exosuit in randomised orders. The lifting tasks included (1) vertical lifts of different weights (15 and 25 kg) onto different platform heights (0.5 m and 1.2 m) and (2) a lateral walk task across 4 m in a confined space while carrying a 39 kg weight. EMG activities of three back muscle groups (longissimus, iliocostalis, and multifidus) were measured and normalised to maximal isometric back extension tasks. The results showed no significant differences in muscle activation between conditions in most lifting tasks, except for a reduction in longissimus muscle activity when using the exosuit during lateral walking. Individual responses varied substantially, with some participants showing reduced muscle activity, while others did not. These findings highlight the challenges in implementing exosuits in reducing back muscle load during military lifting tasks. While passive back-support exosuits may provide benefits to some users, their effectiveness varies among individuals and may be task-dependent.

## 1. Introduction

Musculoskeletal injuries are a significant concern for soldiers, who often perform maintenance and repair work duties on military vehicles. Tasks such as lifting and transporting heavy loads in confined spaces can place excessive strain on the back [1]. These conditions, coupled with awkward posture and poor lifting technique, may increase the risk of both short and long-term musculoskeletal injuries [2]. This can lead to discomfort and poor quality of life, with some soldiers even being at risk of being discharged from duty [3]. In Singapore, back injuries are the most common work-related musculoskeletal disorder, mainly caused by sudden and repetitive overloading, carrying heavy loads, and poor posture [4].

The use of an exoskeleton is a potential solution to prevent low back pain and injuries by reducing the physical load on the wearer’s body via mechanical structures such as elastic bands, springs, and ergonomic designs [5]. Its application has been explored in multiple use cases such as in construction, agriculture, and military tasks [6,7,8]. Compared to active exoskeletons, passive exoskeletons are more practical for operations because they are lightweight, less bulky, and allow a wider range of motion, providing greater mobility for users [9]. These wearable devices assist users by distributing the physical load evenly to reduce the risk of overexertion. To quantify the effectiveness of exoskeletons on muscle activity, surface electromyography (EMG) is frequently employed. Surface EMG is a non-invasive technique that measures electrical activity in muscles, providing valuable insights into muscle activity, fatigue, and workload during physical tasks, making it a critical tool for evaluating the effectiveness of exoskeletons. Studies have shown that passive exoskeletons are capable of reducing lower back muscle activity during heavy lifting [10,11,12,13]. In military settings, the use of back exoskeletons has received positive feedback from soldiers, who believe these devices can enhance their operational performance [14].

Recently, the Auxivo LiftSuit (Auxivo AG, Schwerzenbach, Switzerland), a passive back-support exosuit, has shown the potential to reduce muscle strain and fatigue during lifting tasks [10,12,13,15,16]. Goršič et al. [13] evaluated the effects of the LiftSuit1.1 during lifting and static leaning tasks, revealing a significant reduction in EMG activity of the erector spinae. The findings suggested that the LiftSuit not only reduces mean and peak muscle activity but also helps prevent fatigue and injuries associated with heavy lifting. Similarly, Refai et al. [15] demonstrated notable decreases in the activity of the iliocostalis and longissimus muscles, which are essential for back stability. The newer model, the LiftSuit2.0, has also been shown to reduce muscle activity in the lower back by up to 20% during lifting tasks [10]. Another study examining the LiftSuit2.0 [17] showed reduced muscle fatigue when participants were asked to hold a load for prolonged periods. These results underscore the effectiveness of the Auxivo LiftSuit in alleviating the physical demands placed on the back during lifting tasks. However, the previous studies primarily involved generic lifting tasks with loads under 20 kg. It remains unclear if the LiftSuit2.0 passive back-support exosuit is effective in reducing back muscle activity in specific military lifting tasks with heavier loads.

Thus, the purpose of this study was to compare the back muscle activity when performing military-relevant lifting tasks with and without wearing a passive back-support exosuit. It was hypothesised that the exosuit would be effective in reducing back muscle activity during dynamic lifting activities that involved different loads, lift heights, and lift directions.

## 2. Materials and Methods

### 2.1. Study Design

This study adopted a within-subject, randomised cross-over design. Two experimental sessions were conducted in a laboratory on separate days, with and without the use of a back-support exosuit. Participants were asked to perform various military-relevant lifting tasks with heavy loads. A wireless electromyography (EMG) system was used to evaluate back muscle activity during the lifting tasks.

### 2.2. Back-Support Exosuit

The Auxivo LiftSuit2.0 (Auxivo AG, Schwerzenbach, Switzerland) is a lightweight (~1 kg) passive exosuit that is designed to support the user’s back muscles during lifting activities (Figure 1) [10]. Textile springs, which are located parallel to the back muscles when donned, are stretched when the user leans forward. The elastic force generated is then transferred to the body through its textile materials and attachment points at the chest and thighs. The LiftSuit2.0 is available in multiple sizes and is adjustable to accommodate different users’ thigh, hip, and chest circumferences and torso lengths to ensure proper fit and function. As there were individual differences in height and weight among participants, we assigned the exosuit sizes following the manufacturer’s recommendations. Taller and heavier participants used L/XL sized exosuits, while shorter and lighter individuals used S/M sizes.

### 2.3. Participants

This study was approved by the Nanyang Technological University Institutional Review Board (NTU-IRB, Ref no.: IRB-2023-799). Participants were recruited from a pool of active military personnel, and they had to be (1) males, (2) aged between 21 and 45 years old (inclusive), and (3) currently healthy to undertake the lifting of heavy loads. Potential participants who had a history of any back or leg surgery would be excluded from this study. A total of 10 participants were recruited (Table 1). They were briefed on the protocols and risks involved with this study before signing an informed consent form.

### 2.4. Wireless Surface Electromyography (EMG) System

To evaluate the effectiveness of the back-support exosuit, a wireless surface electromyography (EMG) system (Noraxon MR3, Scottsdale, AZ, USA) was used to record the back muscle activity during the lifting tasks. A key advantage of using this wireless system is that it allows the participants to move freely without being restricted by any wires or cables connecting the EMG sensors to a data logger or a computer. Three lower back muscles were selected for EMG measurements, including the erector spinae (longissimus), erector spinae (iliocostalis), and multifidus. These muscle groups were selected because the erector spinae muscle group is the main contributor to the back extension movement, while the multifidus plays an important role in stabilising the spine during dynamic activities [18,19]. The wireless EMG system samples at 1500 Hz and is incorporated with a Logitech HD Pro Webcam C920, capturing video at 1080 p and 30 frames per second. This setup enables visual identification of key time-points during a movement task, facilitating muscle activity analysis during different phases.

### 2.5. Procedures

Two sessions of experimental tests, with and without wearing the exosuit, were conducted at least 48 h apart to allow sufficient recovery time. Counterbalanced randomisation procedures were performed to determine if a participant would wear the exosuit during the first or second session. At each session, participants completed a series of vertical and lateral lifting tasks, completing the vertical tasks first before proceeding to the lateral task. Participants completed the tasks in full-length pants and safety boots to simulate operational attire.

#### 2.5.1. Skin Preparation

To improve the quality of the recorded electrical signal via the EMG sensors, skin preparation procedures are required to minimise the impedance at the electrode–skin interface. Following the common practice in EMG studies, the skin of the participant was carefully prepped to ensure it was free from extraneous matter such as hair, oils, and dry dermis. This is achieved by cleaning the target muscle areas, shaving off excessive body hairs in these areas, followed by wiping the skin with alcohol pads.

#### 2.5.2. EMG Sensors Placement

EMG sensors were placed on the left and right target muscles (longissimus, iliocostalis, and multifidus) based on the Surface Electromyography for the Non-Invasive Assessment of Muscles European (SENIAM) recommendations [20]. A total of six EMG sensors (three left, three right) were placed on the participants’ lower back. To guide the sensor placement, markings were drawn on the participants’ backs to pinpoint the exact location of each sensor (Figure 2). These markings allow the researcher to place the EMG sensors accurately and consistently across all participants. Adhesive tapes were applied to further secure the EMG sensors to the skin. This step is important to prevent any electrode movement from its original position during the dynamic tests, ensuring the consistency and quality of the EMG signals. To control for environmental factors, this study was conducted in an air-conditioned laboratory where humidity and temperature were kept fairly constant. This environment minimised the likelihood of significant drift in the EMG signals due to participant sweating.

#### 2.5.3. EMG Normalisation Task

The EMG data with and without wearing the exosuit were collected on two separate days. Thus, data normalisation is a vital step to draw meaningful comparisons between different days due to the inherent EMG signal variability [21,22,23]. As this study focused on strenuous lifting activities, a maximum voluntary isometric back extension task was chosen as the reference for EMG normalisation (Figure 3).

The maximum isometric back extension task was performed at the beginning of each test session before the lifting tasks without wearing the Exosuit. Participants stood on a back dynamometer (Takei T.K.K.5402 BACK-D, Takei Scientific Instruments Co., Ltd., Tokyo, Japan), which measured the back extension strength in kg. To control for confounding factors, we standardised the test posture by asking participants to keep their legs straight and hip angle at approximately 120° using a goniometer. This testing posture for maximum isometric back extension has been previously used in other occupations that involve strenuous lifting tasks [24,25]. Participants were instructed to pull the dynamometer handlebar up with their maximal effort using their back muscles for 3 s. A briefing and demonstration were first conducted, with emphasis on generating force from their backs and not their legs. Participants then practice before performing a total of three attempts, with a rest period of 30 s in between. The maximum isometric back extension strength did not differ between the two test days (Exo: 91.1 (20.6) kg; Non-Exo: 88.5 (14.8) kg; *p* = 0.600). The intra-class correlation analysis (ICC = 0.70) also reflected good reliability of the MVC test protocol.

#### 2.5.4. Vertical Lifting Tasks

After completing a self-selected 5 min warm-up and familiarisation with the movements and equipment, participants began their assigned trials. They were asked to lift the load from the ground and place it onto a platform positioned 50 cm in front of them (Figure 4). The vertical lifts included two different loads (15 kg, 25 kg) and two different platform heights (0.5 m, 1.2 m). This was referenced from soldiers’ actual work environment where they are expected to lift maintenance equipment (~15 kg) or vehicle batteries (~25 kg) from the ground onto the vehicle ramp (~0.5 m) or a shelf (~1.2 m). The orders of the load and platform height were randomised among the participants. Each lift was performed three times, making a total of 12 lifts (2 loads × 2 heights × 3 trials) per participant. To minimise fatigue, participants were given at least 2 min to rest in between trials.

For better ecological validity, the research team did not restrict the lifting technique or require participants to complete the tasks within a given period of time. Instead, participants were briefed to lift the loads in any method that they were most familiar and comfortable with. This approach will provide insights into the potential effectiveness of the back-support exosuit in a real-life work environment where individuals may have different lifting techniques and may naturally adjust their movement patterns after donning an exosuit.

#### 2.5.5. Lateral Lifting Task

Upon completion of the vertical lifting tasks, participants proceeded to execute the lateral lifting task. This task required participants to carry a 39 kg vehicle battery laterally across a distance of 4 m within a confined space (height: 1.2 m, width: 1.4 m), designed to simulate the inside dimensions of a military vehicle (Figure 5). During the lateral walk, participants were asked to picture an imaginary roof above them and prompted to keep their head position below its 1.2 m height throughout. Each participant completed a total of three lateral walk trials, with a 3 min break in between each trial. While the research team did not specify the direction of travel, all participants completed the lateral walk task facing the same direction, leading with the left leg.

### 2.6. Data Processing

Key events of the lifting tasks were visually identified from the video recordings and manually marked using the MR3 Noraxon software (MR3.8.30). For the vertical lifts, the key events included the last instant of standing still (*t*_1_), lowest squat position and grabbing the load (*t*_2_), lifting the load and taking one step forward(*t*_3_), and placing the load down (*t*_4_). The lowest squat position was determined visually based on the hip position from the synchronized video recordings. These four timepoints divided the vertical lifts into three distinct phases, comprising the P1: Lowering Phase (*t*_1_ to *t*_2_), P2: Lift and Forward Phase (*t*_2_ to *t*_3_), and P3: End Phase (*t*_3_ to *t*_4_) (Figure 2). For the lateral lifting task, the start of the trial was determined from the instant when the load was off the ground. Likewise, the trial was considered ended when the participant had completed shifting the load across the 4 m distance and the load was fully in contact with the ground. The time to completion (in seconds) of the lateral lift task was used as an indication of operational performance.

Raw EMG data of the six muscles for all trials were processed using Visual 3D in the following manner: (1) band-pass filtered between 20 to 450 Hz, (2) full-wave rectification, and (3) smoothing via root mean square (RMS) of 100 ms. The maximum of the RMS-EMG over each of the three phases of the vertical lifts was calculated. For the lateral lift and the maximal isometric back extension tasks, the maximum of the RMS-EMG of the entire movement trial was computed. Out of the three trials of the maximal isometric back extension, the highest maximal RMS-EMG value was used as a reference for normalisation [13,26]. The maximum RMS-EMG of all lifting tasks was then normalised to this reference value to allow meaningful between-day comparisons. For each lifting condition, the mean of the three normalised RMS-EMG values was calculated.

### 2.7. Statistical Analysis

Statistical analysis was performed using R (version 4.4.1). Data are expressed as mean (standard deviation) unless otherwise stated. For group-level statistical analysis, the significance level was set as *p* < 0.05. Additionally, individual analysis was also performed to provide insight into the variations in responses among the 10 participants.

#### 2.7.1. Vertical Lifting Tasks

As the vertical lifts are largely bilaterally symmetrical, it can be expected that muscle activity would be similar between the left and right sides. A series of paired t-tests were performed and confirmed that there was no significant difference in the normalised RMS-EMG between the two sides in all the vertical lift tasks (*p* > 0.05). To simplify the statistical analysis, therefore, the mean values of the left and right muscles of each participant were calculated to represent the muscle activity for the particular condition.

Previous research showed that the EMG of back muscles changed substantially across different phases of a vertical lift task and that the exosuit was most effective in reducing peak EMG during the upward phase [10]. In the present study, the loading demand during the lowering phase was expected to be much lower than the other two phases because the participants did not carry any external load when lowering the body. The back EMG activities would be higher during the active upward lifting phase than the end phase when the participant unloaded the weight. As such, it is necessary to analyse the EMG activities of the different phases separately rather than taking an average over the entire lifting duration. The assumption of normality was assessed using the Shapiro–Wilk test, and the homogeneity of variance was assessed using Levene’s test. Subsequently, for each muscle, a 2 *×* 3 (Conditions *×* Phases) repeated measures Analysis of Variance (ANOVA) was conducted on the normalised RMS-EMG at each height (0.5 m, 1. 2 m) and load (15 kg, 25 kg) to determine the effects of the exosuit. Post hoc comparison with *Bonferroni* adjustment was applied where appropriate.

#### 2.7.2. Lateral Lifting Task

The lateral lifting task was bilaterally asymmetrical in nature and hence the normalised RMS-EMG of the left and right muscles were analysed separately. For each muscle, a 2 × 2 (Conditions × Sides) repeated measures ANOVA was performed. Additionally, a paired t-test was conducted to evaluate if there was a significant difference in the time to completion between wearing and not wearing the exosuit.

#### 2.7.3. Individual Analysis

For both vertical and lateral lifting tasks, individual normalised RMS-EMG responses of the 10 participants were plotted and analysed to compare conditions with and without the exosuit. To simplify the analysis in the vertical lifting tasks, only the P2: Lift and Forward Phase was analysed, as this was the most strenuous phase where participants lifted the load from the ground towards the platform.

## 3. Results

### 3.1. Vertical Lifting Tasks

Most of the data met the normality assumption. Of the 72 sets of data (3 muscles × 3 phases × 2 heights × 2 loads × 2 conditions), only 4 (5.6%) violated normality (*p* < 0.05). Given that ANOVA is robust to minor violations of normality [27], the analysis proceeded as planned. Across all muscles, loads, and heights, the Phases factor showed a significant effect (Table 2, Figure 6). However, there were no significant effects for the Conditions factor, nor was there any interaction effect between Conditions and Phases. No post hoc analysis was conducted, as the effects of Phases were expected and deemed inconsequential to the impact of exosuit usage.

Individual analyses revealed considerable variations among the 10 participants (Figure 7). One participant (S03) was a responder to exosuit, showing lower normalised %MVC across all muscles and lifting conditions when wearing the exosuit. On the other hand, there was also one non-responder (S05) showing the reverse in all trials. Mixed responses were observed in the other eight participants, varying across different muscles, loads, and lift heights.

### 3.2. Lateral Lifting Task

Paired t-test results revealed no significant difference in the time to completion between exosuit (10.10 ± 2.04 s) and non-exosuit (9.64 ± 2.62 s) conditions (t = −0.66, *p* = 0.53). Individual analysis showed time improvement in 3 out of 10 participants when using the exosuit, as highlighted in green in Figure 8.

As only 1 out of the 12 RMG-EMG datasets for the lateral lifting task (3 muscles × 2 conditions × 2 sides) violated the normality assumption, the ANOVAs were conducted as planned. The mean normalised %MVC of each muscle was generally lower in the exosuit condition than in the non-exosuit condition (Figure 9). However, only the longissimus muscle showed a significant main effect in the Conditions (Table 3). There were no significant differences or interaction effects in the iliocostalis and multifidus muscles. Four participants (S02, 03, 04, and 10) had a reduced normalised %MVC in the exosuit compared with the non-exosuit conditions on both left and right sides (Figure 10). The other six participants did not show a consistently positive response to the exosuit.

## 4. Discussion

This study aimed to evaluate the effectiveness of a back-support exosuit in military-relevant lifting tasks, focusing on both vertical and lateral lifting movements. Contrary to the hypothesis, the exosuit did not significantly reduce back muscle activity across the tasks, with large inter-individual variations in the responses to the exosuit. These results indicate that the back-support exosuit could not provide consistent support across individuals and tasks, highlighting the need to consider user-specific factors for military applications.

### 4.1. Group Responses to Back-Support Exosuit

In the vertical lifting tasks, the presence of a significant phase effect was expected, as muscle activation naturally varies throughout the lifting phases due to different movements. Previous studies [10,13,15,16] have shown reductions in back muscle activity, indicating the effectiveness of back-support exosuits. In this study, however, no significant main effects in Conditions or interaction were observed between Conditions and Phases. These findings contrast with earlier studies investigating the Auxivo LiftSuit’s effectiveness in lifting tasks. One possible explanation is that previous studies only looked at vertical lifting tasks on the same spot [10,13] whereas, in the present study, participants were required to lift the load upwards, move forward, and then place the load onto a platform placed 50 cm in front. Additionally, the loads used in this study were slightly heavier (15 kg and 25 kg) than those in previous studies [10,13]. The loads were chosen to accurately reflect the typical conditions military personnel would encounter in their daily tasks. Given our findings, it is possible that the exosuit is less effective at heavier loads. Future research could explore the load threshold at which the passive back-support exosuit can provide meaningful assistance, to better understand its limitations and optimal use cases.

The use of the back-support exosuit during lateral lifting tasks yielded mixed results across the time performance and neuromuscular measurements. While there was no significant difference in the average time to completion between conditions, it is worth noting that wearing the exosuit led to a marginally longer time. This indicates that the use of the exosuit could have interfered with the movement during the task, resulting in a slower completion time. From a neuromuscular perspective, the exosuit generally reduced mean back muscle activity, but significant differences were only observed in the *longissimus* muscle. Notably, the longissimus in the ipsilateral side of the lead leg demonstrated consistently higher activity. On the opposite side, the iliocostalis and multifidus in the ipsilateral side of the trail leg showed a pattern of higher muscle activity. Comparing our findings with the existing literature is challenging, as, to the authors’ knowledge, no studies have assessed exosuit use during sideways walking with external loads. Previous studies [6,28] examined lifting an external load by performing a trunk rotation while the legs are stationary. Our findings present novel insights into asymmetrical muscle activation patterns during lateral lifting tasks. The observed asymmetries and variable effectiveness across users underscore the importance of designing exoskeletons that can provide targeted support and accommodate individual movement strategies. Currently, the findings indicate that the passive back-support exosuit does not yet provide sufficient support for sideways movement while carrying a 39 kg external load.

### 4.2. Individual Responses to Back-Support Exosuit

In both vertical and lateral lifting tasks, high inter-individual variability in the responses to the exosuit was observed. For example, in the vertical lifting tasks, participant S03 responded positively to the exosuit, as reflected by the consistent lower muscle activity across all conditions, but participant S05 exhibited the opposite trend across all trials. In the lateral lifting tasks, four participants (S02, S03, S04, and S10) showed lower muscle activity on both left and right sides when wearing the exosuit. However, other participants exhibited varied responses, with some even showing higher muscle activation with the exosuit. The high inter-individual variability suggests that the exosuit may benefit certain users (i.e., responders) while being ineffective or even counterproductive for others (non-responders). As such, the effectiveness of the exosuit may depend on individual biomechanics, including posture and lifting technique. We have explored sub-group analyses to identify specific characteristics (e.g., age, height, strength) of those who responded positively to the exosuits. However, due to the limited number of participants, we are unable to confirm whether any demographic or physical characteristics are related to the effectiveness of exosuit use.

To better simulate real-life conditions, the lifting technique was not controlled in the present study. One previous study on the Auxivo Exosuit [10] examined muscle activity during a well-controlled lifting task that required participants to lift at specified joint angles (90 degrees knee flexion, 45 degrees trunk flexion) and paced the movement at 5 s per cycle. In reality, soldiers will likely adopt different lifting techniques and complete the tasks at different speeds. Gorsic et al. [13] observed that when participants had the autonomy to choose lifting techniques while wearing the exosuit, they preferred using the stooped lifting technique. The authors postulated that the exosuit’s stiffness around the lower back and hips may have made squatting uncomfortable, leading participants to naturally lean into the exosuit for support when lifting. In the present study, we observed a wide range of lifting techniques used by our participants (Figure 11). The different biomechanical loading on back muscles among the various postures and techniques may explain why we observe diverse responses to the exosuit. To optimise the design of military assistive devices, it is critical to consider inter-individual differences in factors such as physical characteristics, strength, posture, and movement techniques.

Refai et al. [15] acknowledge that the Auxivo may be uncomfortable for some participants. However, these earlier studies were using the first version, and the LiftSuit2.0 may have improved slightly by having a more comfortable leg cuff and less stiffness around the hip [10]. The substantial variability in participants’ physical characteristics (Table 1), specifically the larger standard deviation in body mass compared to height, may have also contributed to the varied individual responses to the use of the exosuit during lifting tasks. Although subjective feedback on the use of the exosuit was not collected in this study, potential discomfort may have contributed to the increased task completion time observed in 70% of participants during the lateral lifting task.

Pertaining to work efficiency, only 3 in 10 participants completed the lateral lifting task faster when wearing the exosuit, while the other 7 showed a slower completion time. These performance outcomes, in alignment with the muscle activity findings, suggest that the exosuit can only benefit some individuals, but it does not provide a consistent advantage for all users. The mixed findings raise questions about the real-world applicability of the exosuit in military settings, highlighting the need to better understand the factors influencing individual responses to exosuits. Future studies could explore the effects of prolonged exosuit use to determine whether adaptation over time leads to improved performance and reduced muscle activity.

### 4.3. Limitations

Several limitations of this study should be acknowledged. First, the sample size was limited to 10 participants, which likely compromised the statistical power of the results. We are also unable to consider lifting heights and loads in the statistical tests because more complex analysis will require a larger sample size. Future work should recruit more participants such that the effects of exoskeleton use can be examined statistically together with other factors. Second, the findings are specific to the Auxivo LiftSuit2.0 and the lifting tasks used in this study. Therefore, these results should not be directly generalised to other types of exoskeletons or lifting tasks. Third, this study focused on EMG data of the back muscles, without incorporating kinematic or kinetic measurements. To provide a more comprehensive assessment of the exosuit’s effectiveness, future research should include additional variables such as joint angles and perceived exertion, which have been examined in other exoskeleton studies [10,13]. Evaluating other muscle groups involved in lifting tasks, such as abdominal and leg muscles, could offer additional insights into the benefits and limitations of the exosuit. Lastly, while muscle activity can inform the loading intensity during dynamic tasks, we should be mindful of not over-relying on EMG results as a surrogate for injury risk. Longitudinal tracking of whether exosuit use can reduce injury occurrence at the workplace is necessary to confirm its long-term effectiveness.

## 5. Conclusions

This study showed that the use of the back-support exosuit did not reduce back muscle activation in most military-relevant lifting tasks, except for the longissimus muscle activity during lateral walking while carrying a 39 kg load. The inter-individual responses varied substantially, with some participants showing reduced muscle activity, while others did not. These findings suggest that, while a back-support exosuit may reduce back muscle activation for some individuals, it does not guarantee load reduction or performance improvements across all users. As such, it is important to account for individual variability and task-specific requirements when implementing exosuits in military settings.

## Figures and Tables

**Figure 1 sensors-25-03211-f001:**
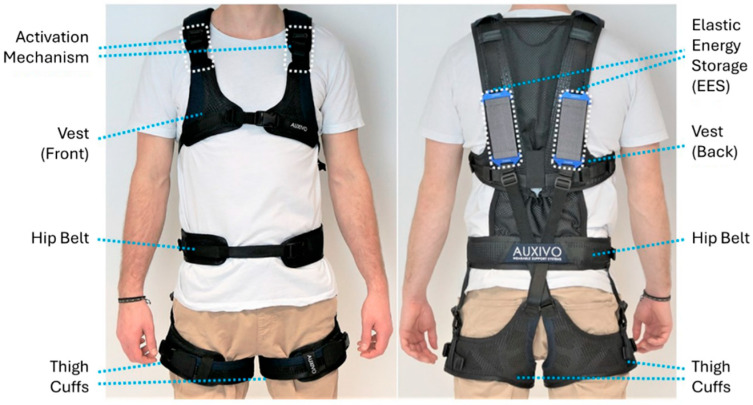
Key elements of the Auxivo LiftSuit 2.0 back-support exosuit from the front and back views.

**Figure 2 sensors-25-03211-f002:**
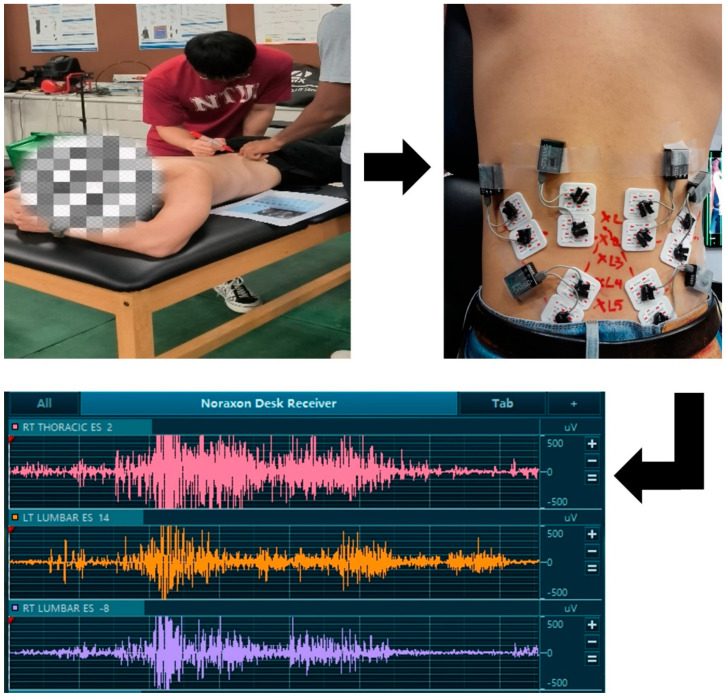
Markings were drawn on the participant’s back to guide EMG sensor placement for wireless data acquisition of muscle activity during lifting tasks.

**Figure 3 sensors-25-03211-f003:**
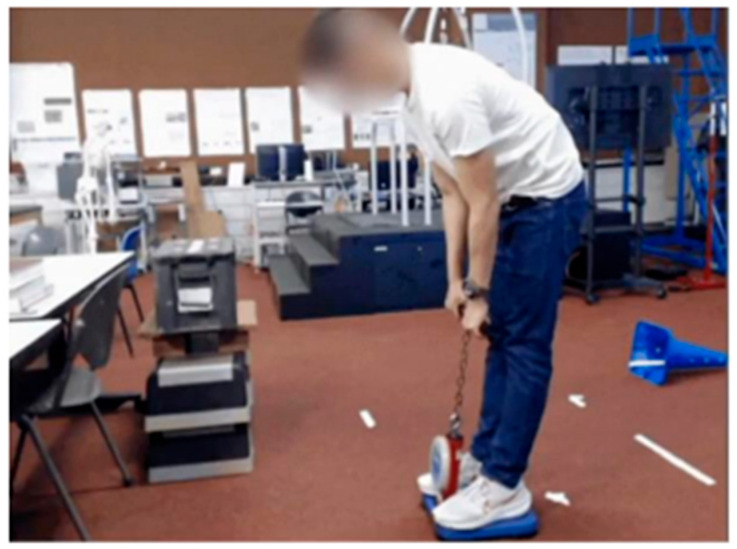
A maximal isometric back extension task was used as a reference for normalisation of EMG data to allow between-day comparisons.

**Figure 4 sensors-25-03211-f004:**
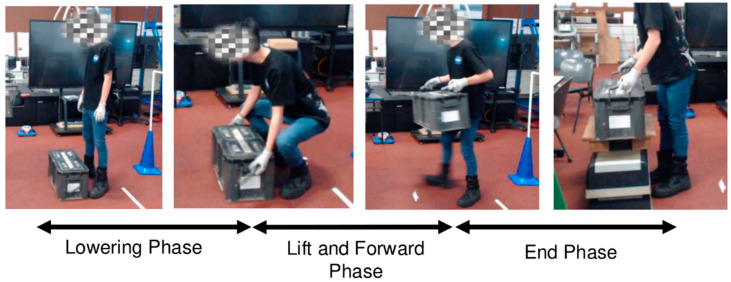
Participants were tasked to lift a load from the ground and place it onto a raised platform positioned 50 cm in front of them.

**Figure 5 sensors-25-03211-f005:**
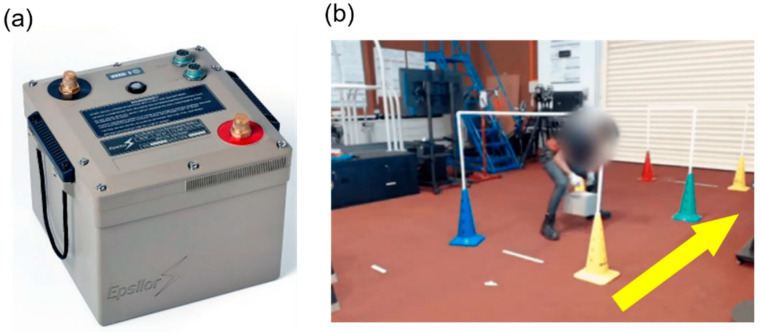
(**a**) A vehicle battery weighing 39 kg. (**b**) A participant executing the lateral walk task in a confined space (1.2 m × 1.4 m × 4 m) while carrying the load. The yellow arrow indicates the direction of travel.

**Figure 6 sensors-25-03211-f006:**
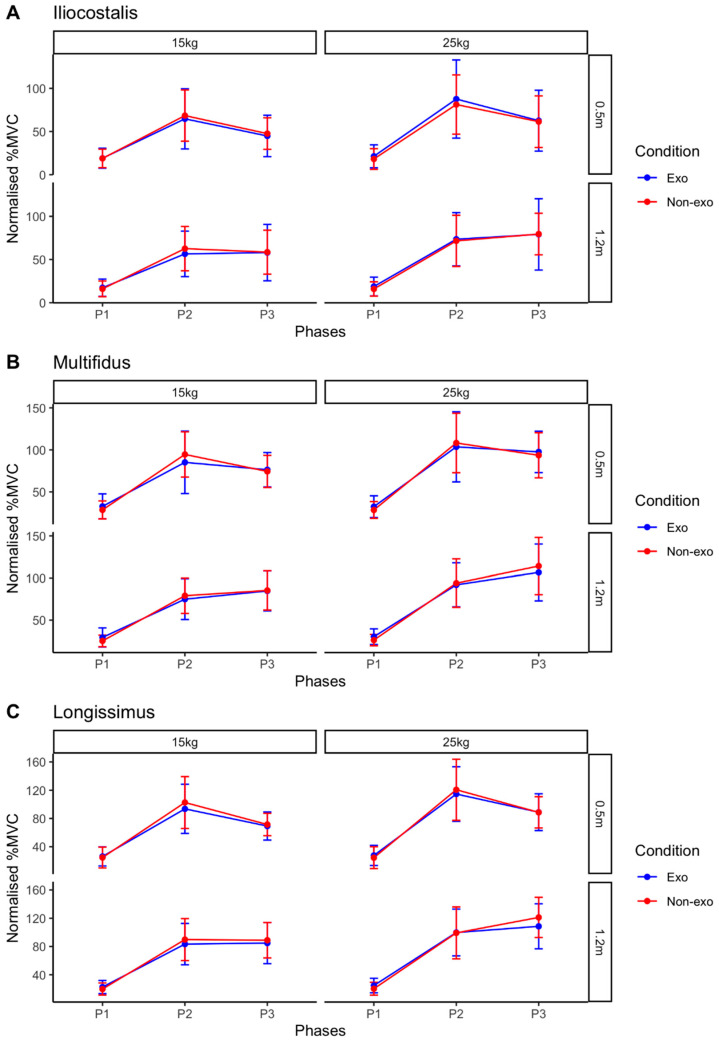
Normalised %MVC averaged across all participants for the (**A**) iliocostalis, (**B**) multifidus, and (**C**) longissimus muscles during vertical lifting tasks. P1: Lowering Phase; P2: Lift and Forward Phase; and P3: End Phase. Error bars represent 1 standard deviation. %MVC is displayed across different loads (columns) and lifting heights (rows) under two conditions: exosuit-assisted (blue) and non-exosuit (red) conditions for each muscle.

**Figure 7 sensors-25-03211-f007:**
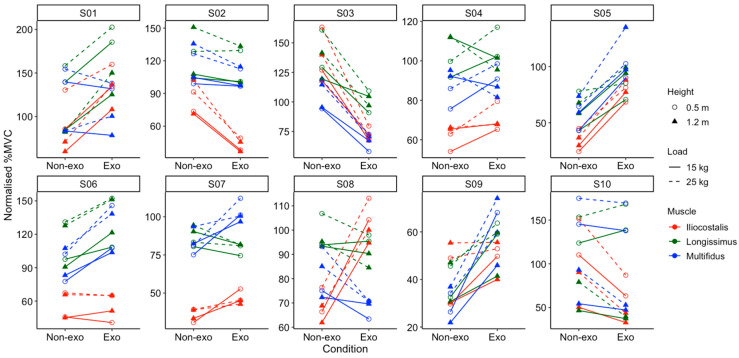
Normalised %MVC in all participants (S01–S10) across different lifting heights (0.5 m represented by circles; 1.2 m represented by triangles), loads (15 kg represented by solid lines; 25 kg represented by dashed lines), and muscles (iliocostalis in red; longissimus in green; multifidus in blue) in the vertical lifting tasks. Exo and Non-Exo refer to the use of an exosuit and no exosuit, respectively.

**Figure 8 sensors-25-03211-f008:**
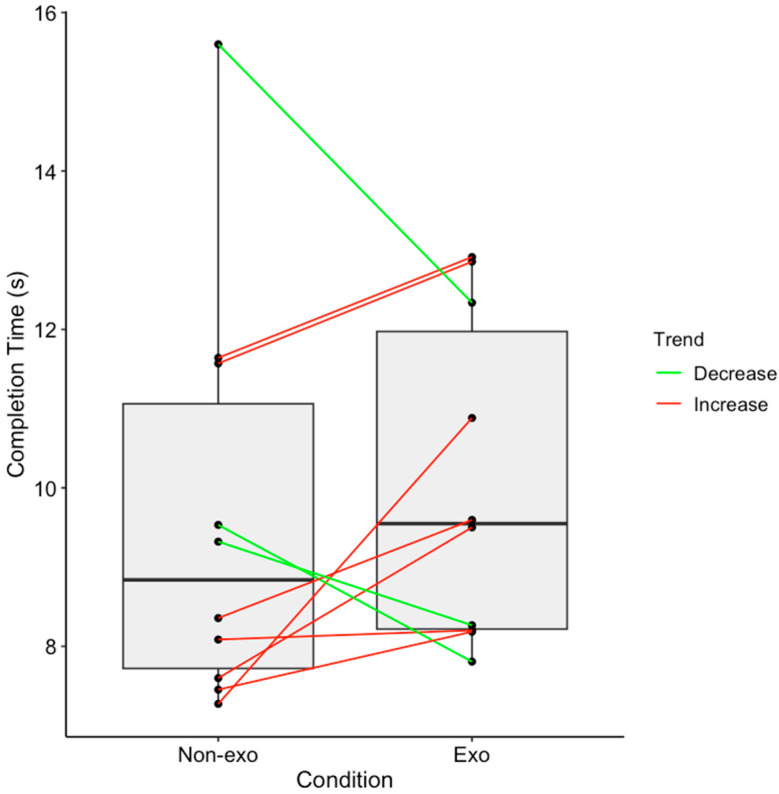
Time to completion for lateral lifting tasks, with each dot representing a single participant. Trend lines shown indicate comparative performance between conditions: green lines indicate faster completion time with the exosuit compared to the non-exosuit condition, while red lines indicate the opposite.

**Figure 9 sensors-25-03211-f009:**
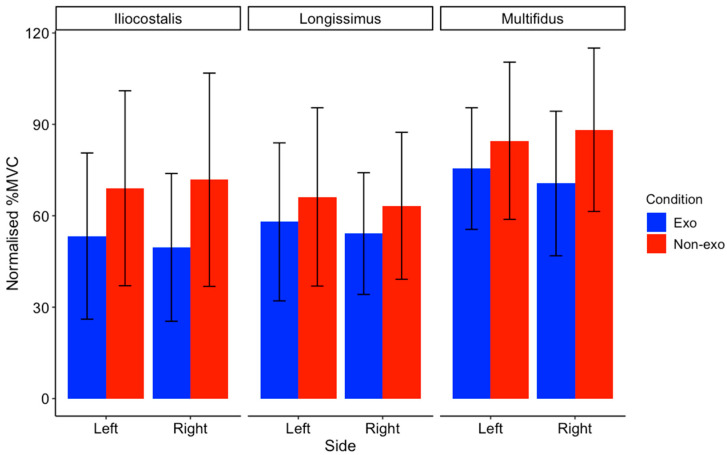
Group mean (SD) of normalised %MVC for the iliocostalis, multifidus, and longissimus muscles during lateral lifting tasks, separated by side (left in blue; right in red).

**Figure 10 sensors-25-03211-f010:**
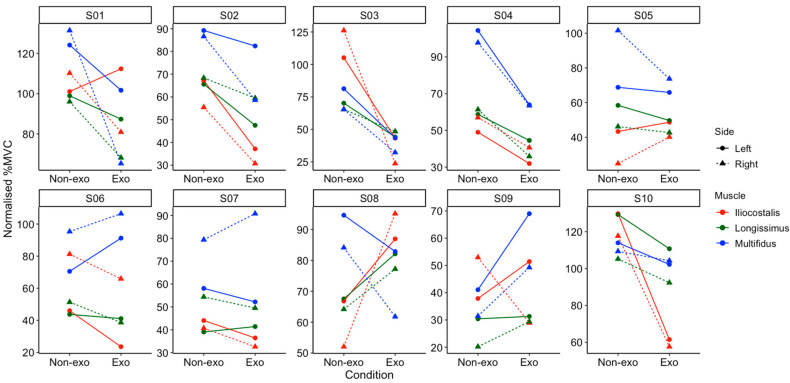
Normalised %MVC in all participants (S01–S10) for each muscle (iliocostalis in red; longissimus in green; multifidus in blue) and sides (left represented by circle; right represented by triangle) in the lateral lifting tasks.

**Figure 11 sensors-25-03211-f011:**
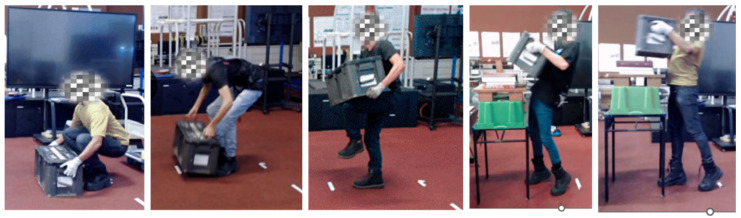
Participants used a wide range of techniques to perform lifting tasks.

**Table 1 sensors-25-03211-t001:** Physical characteristics of 10 male participants.

Participants	Age (Years)	Body Mass (kg)	Height (cm)
S01	23	60.6	173
S02	28	72.5	181
S03	26	58.5	179
S04	43	99.0	185
S05	27	83.2	172
S06	37	67.5	170
S07	37	92.3	177
S08	45	94.0	179
S09	30	50.9	179
S10	29	48.0	160
Mean (SD)	32.1 (7.0)	72.6 (17.6)	175.5 (6.6)
Range	23–45	48.0–94.0	160–185

**Table 2 sensors-25-03211-t002:** The mean ± standard deviation of normalised muscle activity and the two-way ANOVA results from different heights, loads, and muscles in the vertical lifting tasks.

Muscle	Height (m)	Load (kg)	Phases						Conditions		Phases		Interaction	
			P1: Lowering		P2: Lift and Forward		P3: End							
			Non-Exo (% MVC)	Exo (% MVC)	Non-Exo (% MVC)	Exo (% MVC)	Non-Exo (% MVC)	Exo (% MVC)	*p*	$\eta_{p}^{2}$	*p*	$\eta_{p}^{2}$	*p*	$\eta_{p}^{2}$
Iliocostalis	0.5	15	19.03 ± 11.60	18.85 ± 10.67	64.71 ± 34.87	68.46 ± 29.72	44.86 ± 23.95	47.58 ± 18.31	0.773	0.01	<0.001	0.797	0.902	0.011
		25	21.32 ± 13.29	18.19 ± 12.00	87.63 ± 45.31	81.26 ± 34.36	62.58 ± 35.29	61.42 ± 29.87	0.705	0.017	<0.001	0.761	0.868	0.016
	1.2	15	17.29 ± 10.05	16.02 ± 9.14	56.52 ± 26.30	62.64 ± 25.72	58.03 ± 32.61	58.56 ± 25.46	0.802	0.007	<0.001	0.752	0.633	0.050
		25	18.74 ± 10.83	15.88 ± 8.31	73.54 ± 30.83	71.60 ± 29.73	79.13 ± 41.29	79.53 ± 24.04	0.873	0.003	<0.001	0.859	0.949	0.006
Multifidus	0.5	15	32.71 ± 14.84	28.51 ± 10.67	85.13 ± 37.17	94.57 ± 26.93	76.33 ± 20.54	74.33 ± 19.14	0.865	0.003	<0.001	0.817	0.275	0.134
		25	32.37 ± 12.98	28.39 ± 10.07	103.60 ± 41.76	108.20 ± 35.37	97.57 ± 24.61	93.54 ± 26.84	0.881	0.003	<0.001	0.810	0.628	0.050
	1.2	15	29.34 ± 11.35	25.21 ± 6.97	74.89 ± 24.26	79.02 ± 21.05	84.70 ± 23.77	85.28 ± 23.36	0.970	<0.001	<0.001	0.785	0.598	0.056
		25	30.20 ± 9.31	26.17 ± 6.69	91.89 ± 26.17	93.90 ± 28.82	106.58 ± 33.72	114.15 ± 33.97	0.806	0.007	<0.001	0.825	0.625	0.051
Longissimus	0.5	15	26.29 ± 13.48	24.82 ± 14.66	93.61 ± 34.75	102.63 ± 36.65	69.31 ± 19.93	71.51 ± 15.86	0.463	0.061	<0.001	0.828	0.367	0.105
		25	27.76 ± 14.20	24.56 ± 15.42	114.57 ± 38.57	120.64 ± 43.18	88.88 ± 26.01	88.68 ± 22.13	0.892	0.002	<0.001	0.829	0.440	0.087
	1.2	15	22.64 ± 9.29	19.83 ± 8.60	83.40 ± 29.18	89.81 ± 29.64	84.85 ± 29.17	88.93 ± 25.03	0.521	0.047	<0.001	0.774	0.412	0.094
		25	24.84 ± 10.34	20.27 ± 9.15	99.80 ± 33.12	99.20 ± 36.69	108.53 ± 31.85	121.13 ± 28.36	0.658	0.023	<0.001	0.832	0.322	0.118

Note: $\eta_{p}^{2}$ indicates partial eta-squared. Exo and Non-Exo refer to the use of an exosuit and no exosuit, respectively. % MVC indicates the percentage of the maximal voluntary contraction.

**Table 3 sensors-25-03211-t003:** The mean ± standard deviation of normalised muscle activity and the two-way ANOVA results from different muscles in the lateral lifting tasks.

Muscle	Conditions				Conditions		Sides		Interaction	
	Non-Exo		Exo							
	Left (% MVC)	Right (% MVC)	Left (% MVC)	Right (% MVC)	*p*	$\eta_{p}^{2}$	*p*	$\eta_{p}^{2}$	*p*	$\eta_{p}^{2}$
Iliocostalis	69.03 ± 32.00	71.83 ± 35.00	53.33 ± 27.28	49.63 ± 24.26	0.105	0.265	0.926	0.001	0.408	0.077
Longissimus	66.19 ± 29.27	63.27 ± 24.13	58.00 ± 25.93	54.16 ± 19.98	0.045 *	0.376	0.290	0.123	0.771	0.010
Multifidus	84.62 ± 25.81	88.23 ± 26.82	75.50 ± 19.97	70.59 ± 23.73	0.091	0.284	0.909	0.002	0.172	0.197

Note: $\eta_{p}^{2}$ indicates partial eta-squared. Exo and Non-Exo refer to the use of an exosuit and no exosuit, respectively. % MVC indicates the percentage of the maximal voluntary contraction. * *p* < 0.05.

## Data Availability

The data are available for download at the NIE Data Repository.

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
