# Peer review of "Biomechanical Effects of a Passive Back-Support Exosuit During Simulated Military Lifting Tasks—An EMG Study"

_sensors, 2025, doi:10.3390/s25103211_

Round 1

Reviewer 1 Report

Comments and Suggestions for Authors

Thank you for inviting me to review this manuscript titled 'Biomechanical effects of a passive back-support exosuit during simulated military lifting tasks – an EMG study.' The authors present a well-written manuscript that investigates the impact of a passive back-assist exosuit on low back muscle activity during simulated military lifting tasks. The data collection appears to be consistent with previous studies. However, there are several points that warrant further clarification and improvement:

  1. While the study emphasizes military-relevant tasks, there is insufficient information about the participants' military background. Were the participants active military personnel or veterans? Clarifying this aspect is important for contextualizing the relevance and generalizability of the findings.
  2. The rationale for selecting 15 kg and 25 kg loads, as well as the chosen lifting heights, should be provided. Explaining how these parameters relate to typical military lifting scenarios would strengthen the study's practical implications.
  3. It is recommended that the Method section be reorganized to follow the time sequence of the testing procedures. This would enhance clarity and reader comprehension. For instance, the MVIC testing should be included as part of the procedural flow.
  4. When identifying movement phases and events, please clarify how the lowest squat position was determined—was it based on hip position or joint angles from the video recording?
  5. The current statistical approach requires revision. The rationale for comparing different task phases is not well-supported. Instead, based on the study's aims and introduction, it would be more appropriate to compare the effects of load and lifting height between exosuit and no exosuit conditions within each phase (2 by 2 by 2).

Author Response

Reviewer 1

Thank you for inviting me to review this manuscript titled 'Biomechanical effects of a passive back-support exosuit during simulated military lifting tasks – an EMG study.' The authors present a well-written manuscript that investigates the impact of a passive back-assist exosuit on low back muscle activity during simulated military lifting tasks. The data collection appears to be consistent with previous studies. However, there are several points that warrant further clarification and improvement:

[Comment 1] While the study emphasizes military-relevant tasks, there is insufficient information about the participants' military background. Were the participants active military personnel or veterans? Clarifying this aspect is important for contextualizing the relevance and generalizability of the findings.

>> Response: The participants were active military personnel. We have clarified this in the Methods section:

“Participants were recruited from a pool of active military personnel, and they had to be: 1) males, 2) aged between 21 and 45 years old (inclusive), and 3) currently healthy to undertake the lifting of heavy loads.”

[Comment 2] The rationale for selecting 15 kg and 25 kg loads, as well as the chosen lifting heights, should be provided. Explaining how these parameters relate to typical military lifting scenarios would strengthen the study's practical implications.

>> Response: We concur that it is important to clearly explain the rationale of the chosen lift weight and heights to strengthen our study’s practice implications. Please refer to the additional information under section 2.5.4 Vertical Lifting Tasks.

“The vertical lifts included two different loads (15 kg, 25 kg) and two different platform heights (0.5 m, 1.2 m). This was referenced from soldiers’ actual work environment where they are expected to lift maintenance equipment (~15 kg) or vehicle batteries (~25 kg) from the ground onto the vehicle ramp (~0.5 m) or a shelf (~1.2 m).”

[Comment 3] It is recommended that the Method section be reorganized to follow the time sequence of the testing procedures. This would enhance clarity and reader comprehension. For instance, the MVIC testing should be included as part of the procedural flow.

>> Response: Thank you for your suggestion. We have re-organised the Methods section following the time sequence of the testing procedures for better clarity and flow. The MVIC test now comes first (section 2.4.3) before the vertical (section 2.5.4) and lateral (section 2.5.5) lifting tasks. The figure numbers are also re-arranged to fit the new flow.

[Comment 4] When identifying movement phases and events, please clarify how the lowest squat position was determined—was it based on hip position or joint angles from the video recording?

>> Response: We have clarified under Methods that:

“The lowest squat position was determined visually based on the hip position from the synchronized video recordings.”

[Comment 5] The current statistical approach requires revision. The rationale for comparing different task phases is not well-supported. Instead, based on the study's aims and introduction, it would be more appropriate to compare the effects of load and lifting height between exosuit and no exosuit conditions within each phase (2 by 2 by 2).

>> Response: We noted your point that the rationale for comparing different phases of the vertical lifting tasks were not sufficiently explained in our previous manuscript. We have now elaborated on the need to analyse each phase separately, making reference to the literature on lifting biomechanics [10]. The added text reads:

“Previous research showed that the EMG of back muscles changed substantially across different phases of a vertical lift task and that exosuit was most effective in reducing peak EMG during the upward phase [10]. In the present study, the loading demand during the lowering phase was expected to be much lower than the other two phases because the participants did not carry any external load when lowering the body. The back EMG activities would be higher during the active upward lifting phase than the end phase when the participant unloaded the weight. As such, it is necessary to analyse the EMG activities of the different phases separately rather than taking an average over the entire lifting duration.”

We have also considered your suggestion of 2 by 2 by 2 analyses but the small sample size (n = 10) would limit our ability to interpret the statistical results meaningfully. As the main purpose of this study was to compare the back muscle activity with and without wearing a passive back-support exosuit, we opt to exclude the height and load factors in the ANOVA. This will allow us to focus on the main comparison (with and without exosuit). We have acknowledged our limitation in statistical analysis in Section 4.3, and recommended future research to consider the height and load factors together with exosuits.

“First, the sample size was limited to 10 participants, which likely compromised the statistical power of the results. We are also unable to consider lifting heights and loads in the statistical tests because more complex analysis will require a larger sample size. Future work should recruit more participants such that the effects of exoskeleton use can be examined statistically together with other factors.”

Reviewer 2 Report

Comments and Suggestions for Authors

The study addresses an under-researched and practical problem—musculoskeletal injury prevention in military personnel—using a passive exosuit, which is a promising yet underexplored technology. The randomized cross-over design effectively minimizes order effects, and the use of EMG measurements provides objective biomechanical insights. The simulated military lifting tasks (e.g., box lifts, sandbag carries) enhance the ecological validity of the findings.

The small sample size (n=12) and homogeneity of participants (age, fitness level) may restrict the applicability of results to broader military populations. Key methodological details (e.g., EMG normalization protocols, sensor placement accuracy) are insufficiently described, raising concerns about reproducibility. The paper fails to adequately explore the root causes of inter-individual variability in exosuit effectiveness, which is critical for optimizing device design.

EMG Normalization: What specific normalization protocol was used (e.g., MVCs, submaximal contractions)? How were potential confounding factors (e.g., electrode drift) addressed? Were adjustments made to the exosuit for individual anthropometric differences? If so, how might this variability impact the results? What factors (e.g., lifting technique, muscle strength asymmetry) might explain the high variability in EMG reductions across participants? Why did the exosuit show no significant benefit in box lifts but modest benefits in sandbag carries? Could task biomechanics (e.g., load distribution, posture) explain this disparity?

Suggested Revisions

  • Expand the discussion to address the mechanistic basis of inter-individual variability (e.g., biomechanical modeling, subgroup analysis).
  • Clarify methodological details (e.g., EMG protocols, exosuit adjustments) to improve reproducibility.
  • Discuss limitations more critically, including the small sample size and reliance on EMG as a surrogate for injury risk.

Author Response

[Comment 1] The study addresses an under-researched and practical problem—musculoskeletal injury prevention in military personnel—using a passive exosuit, which is a promising yet underexplored technology. The randomized cross-over design effectively minimizes order effects, and the use of EMG measurements provides objective biomechanical insights. The simulated military lifting tasks (e.g., box lifts, sandbag carries) enhance the ecological validity of the findings.

>> Response: Thank you for the positive feedback on the novelty, study design and ecological validity of our work.

[Comment 2] The small sample size (n=12) and homogeneity of participants (age, fitness level) may restrict the applicability of results to broader military populations. Key methodological details (e.g., EMG normalization protocols, sensor placement accuracy) are insufficiently described, raising concerns about reproducibility. The paper fails to adequately explore the root causes of inter-individual variability in exosuit effectiveness, which is critical for optimizing device design.

>> Response: We acknowledge that the previous version did not sufficiently describe the EMG procedures and discuss the critical issue on inter-individual variability. Together with comments from Reviewer 1, we have re-organised and improved the Methods section, including the EMG normalization protocols and sensor placement accuracy. We have also elaborated on the discussion surrounding small sample size and inter-individual variability, including adding a new Figure 11 to illustrate the different lifting techniques used. Please see our specific responses to your “Suggested Revision” below. Thank you for your helpful feedback.

[Comment 3] EMG Normalization: What specific normalization protocol was used (e.g., MVCs, submaximal contractions)? How were potential confounding factors (e.g., electrode drift) addressed? Were adjustments made to the exosuit for individual anthropometric differences? If so, how might this variability impact the results? What factors (e.g., lifting technique, muscle strength asymmetry) might explain the high variability in EMG reductions across participants? Why did the exosuit show no significant benefit in box lifts but modest benefits in sandbag carries? Could task biomechanics (e.g., load distribution, posture) explain this disparity?

>> Response: We apologise that the EMG protocols were not sufficiently described previously. Please see our summary of revisions below:

  1. EMG Protocol - We used maximal isometric contraction as MVC protocol. As suggested by Reviewer 1, we have moved the EMG Normalisation section earlier to follow the time sequence of the testing procedures. We provided more information on how to standardize the testing postures, making reference to previous studies using the same test protocols for occupations that involve heavy lifting tasks. We did not observe any obvious baseline drift in the EMG signal. Neverthelss, some measures have been taken to minimise drift: (i) Conducting the experiment in an air-conditioned environment to minimise sweating which would affect skin-electrode impedance; (ii) Use medical tapes to securely attach the electrodes on the skin, preventing them electrode movement during dynamic tests. To check the reliability of the MCV protocol, we performed a t-test (p = 0.600) and calculated the intraclass correlation coefficient (ICC = 0.70) for the maximal isometric back extension strength between the two test days. Please refer to the manuscript and our specific responses below for more detailed information.

  1. Adjustments to Exosuit - The back-support exosuits are available in multiple sizes and assigned based on size recommendations from the Auxivo Liftsuit 2.0 user guide (https://www.materialshandling.com.au/wp-content/uploads/Manual-LiftSuit-2.0.1.pdf). We have now added more information in the manuscript. Please refer to the manuscript and our specific responses below for more detailed information.

  1. Inter-Individual Variability – We have expanded our discussion on potential factors (e.g., lifting technique, strength) that might explain the high variability in EMG results across participants. We have also added a new Figure 11 to illustrate the different postures and techniques employed by the participants. Please refer to the manuscript and our specific responses below for more detailed information.

  1. Lateral vs Vertical Lifting Tasks – Based on our findings, the exosuit did not work much better in the lateral task (carrying a 39-kg load) than the vertical lifting tasks. On a group level, significant difference was only noted in one muscle in the lateral task. We have since expanded our discussion on the differences in individual techniques which might have affected the EMG results. Please refer to the manuscript and our specific responses below for more detailed information.

Suggested Revisions

[Comment 4] Expand the discussion to address the mechanistic basis of inter-individual variability (e.g., biomechanical modeling, subgroup analysis).

>> Response: We have expanded the discussion on inter-individual variability, including a new Figure 11 to illustrate the different techniques employed by the participants. To enhance the transparency of our subgroup analyses, we revised Table 1 to show the physical characteristics of individual participants rather than simply reporting mean (SD, range) data. Due to the small sample size, we are unable to identify common characteristics among those who responded positively to the exosuit. The changes in text are:

Table 1. Physical Characteristics of 10 Male Participants

Participants

Age (years)

Body Mass (kg)

Height (cm)

S01

23

60.6

173

S02

28

72.5

181

S03

26

58.5

179

S04

43

99.0

185

S05

27

83.2

172

S06

37

67.5

170

S07

37

92.3

177

S08

45

94.0

179

S09

30

50.9

179

S10

29

48.0

160

Mean (SD)

32.1 (7.0)

72.6 (17.6)

175.5 (6.6)

Range

23 - 45

48.0 – 94.0

160 - 185

“We have explored sub-group analyses to identify specific characteristics (e.g. age, height, strength) of those who responded positively to the exosuits. However, due to the limited number of participants, we are unable to confirm whether any demographic or physical characteristics are related to the effectiveness of exosuit use.” 

“In the present study, we observed a wide range of lifting techniques used by our participants (Figure 11). The different biomechanical loading on back muscles among the various postures and techniques may explain why we observe diverse responses to the exosuit. To optimise the design of military assistive devices, it is critical to consider inter-individual differences in factors such as physical characteristics, strength, posture and movement techniques.”

Figure 11. Participants used a wide range of techniques to perform lifting tasks.

[Comment 5] Clarify methodological details (e.g., EMG protocols, exosuit adjustments) to improve reproducibility.

>> Response: We have added more details in the Methods to improve reproducibility, including those of EMG protocols and exosuit adjustments. The changes are:

“The LiftSuit2.0 is available in multiple sizes and is adjustable to accommodate different users’ thigh, hip and chest circumferences and torso lengths to ensure proper fit and function. As there were individual differences in height and weight among participants, we assigned the Exosuit sizes following the manufacturer’s recommendations. Taller and heavier participants used L/XL sized exosuits while shorter and lighter individuals used S/M sizes. “

“To guide the sensor placement, markings were drawn on the participants’ backs to pinpoint the exact location of each sensor (Figure 2). These markings allow the researcher to place the EMG sensors accurately and consistently across all participants. Adhesive tapes were applied to further secure the EMG sensors to the skin. This step is important to prevent any electrode movement from its original position during the dynamic tests, ensuring the consistency and quality of the EMG signals. To control for environmental factors, the study was conducted in an air-conditioned laboratory where humidity and temperature were kept fairly constant. This environment minimised the likelihood of significant drift in the EMG signals due to participant sweating.”

“The maximum isometric back extension task was performed at the beginning of each test session before the lifting tasks without wearing the Exosuit. Participants stood on a back dynamometer (Takei T.K.K.5402 BACK-D, Takei Scientific Instruments Co., Ltd, Tokyo, Japan) which measured the back extension strength in kg. To control for confounding factors, we standardised the test posture by asking participants to keep their legs straight and hip angle at approximately 120° using a goniometer. This testing posture for maximum isometric back extension has been previously used in other occupations that involve strenuous lifting tasks [24-25].”

“The maximum isometric back extension strength did not differ between the two test days (Exo: 91.1 (20.6) kg, Non-Exo: 88.5 (14.8) kg, p = 0.600). The intra-class correlation analysis (ICC = 0.70) also reflected good reliability of the MVC test protocol.”

[Comment 6] Discuss limitations more critically, including the small sample size and reliance on EMG as a surrogate for injury risk.

>> Response: We have revised our limitation with more critical thoughts; thank you for your suggestions. The changes are:

“First, the sample size was limited to 10 participants, which likely compromised the statistical power of the results. We are also unable to consider lifting heights and loads in the statistical tests because more complex analysis will require a larger sample size. Future work should recruit more participants such that the effects of exoskeleton use can be examined statistically together with other factors.”

“Lastly, while muscle activity can inform the loading intensity during dynamic tasks, we should be mindful of not over-relying on EMG results as a surrogate for injury risk. Longitudinal tracking of whether exosuit use can reduce injury occurrence at the workplace is necessary to confirm its long-term effectiveness.”

Round 2

Reviewer 1 Report

Comments and Suggestions for Authors

Thank you for the revisions to the manuscript. There are no more comments.